Off the scale: a new species of fish-scale gecko (Squamata: Gekkonidae: Geckolepis) with exceptionally large scales

Scherz Mark D. mark.scherz@gmail.com 1
Daza Juan D. 2
Köhler Jörn 3
Vences Miguel 4
Glaw Frank 1
1 Sektion Herpetologie, Zoologische Staatssammlung München (ZSM-SNSB) , Munich , Germany
2 Department of Biological Sciences, Sam Houston State University , Huntsville , TX , United States
3 Hessisches Landesmuseum Darmstadt , Darmstadt , Germany
4 Zoologisches Institut, Technische Universität Braunschweig , Braunschweig , Germany
Hrbek Tomas
Electronic publication date: 2017 Feb 7
Publication date: 2017
Volume: 5
Electronic Location ID: e2955
Received 2016 Oct 26; Accepted 2017 Jan 3
Copyright: ©2017 Scherz et al.
Copyright year: 2017
Copyright holder: Scherz et al.
License: This is an open access article distributed under the terms of the Creative Commons Attribution License, which permits unrestricted use, distribution, reproduction and adaptation in any medium and for any purpose provided that it is properly attributed. For attribution, the original author(s), title, publication source (PeerJ) and either DOI or URL of the article must be cited.
License URL: https://creativecommons.org/licenses/by/4.0/

Keywords: Geckolepis megalepis sp. nov., Madagascar, Micro-Computed Tomography, Morphology, Integrative taxonomy, Ankarana, Anti-predator defence, Osteology

Funding: Volkswagen Foundation European Association of Zoos and Aquaria (EAZA) Act for Nature Department of Biological Sciences at Sam Houston State University This research was partly funded by the Volkswagen Foundation, the European Association of Zoos and Aquaria (EAZA), and Act for Nature. JD Daza is funded by The Department of Biological Sciences at Sam Houston State University. There was no additional external funding received for this study. The funders had no role in study design, data collection and analysis, decision to publish, or preparation of the manuscript.

==============================
The gecko genus Geckolepis, endemic to Madagascar and the Comoro archipelago, is taxonomically challenging. One reason is its members ability to autotomize a large portion of their scales when grasped or touched, most likely to escape predation. Based on an integrative taxonomic approach including external morphology, morphometrics, genetics, pholidosis, and osteology, we here describe the first new species from this genus in 75 years: Geckolepis megalepis sp. nov. from the limestone karst of Ankarana in northern Madagascar. The new species has the largest known body scales of any gecko (both relatively and absolutely), which come off with exceptional ease. We provide a detailed description of the skeleton of the genus Geckolepis based on micro-Computed Tomography (micro-CT) analysis of the new species, the holotype of G. maculata, the recently resurrected G. humbloti, and a specimen belonging to an operational taxonomic unit (OTU) recently suggested to represent G. maculata. Geckolepis is characterized by highly mineralized, imbricated scales, paired frontals, and unfused subolfactory processes of the frontals, among other features. We identify diagnostic characters in the osteology of these geckos that help define our new species and show that the OTU assigned to G. maculata is probably not conspecific with it, leaving the taxonomic identity of this species unclear. We discuss possible reasons for the extremely enlarged scales of G. megalepis in the context of an anti-predator defence mechanism, and the future of Geckolepis taxonomy.

Introduction

The genus Geckolepis Grandidier, 1867, endemic to Madagascar and the Comoros, comprises a complex of species that have proven particularly difficult to delimit (Köhler et al., 2009). Known as fish-scale geckos, they have unusually large, imbricate scales and are renowned for their ability to discard a large portion of their integument with extreme ease as a defence mechanism (Gardner & Jasper, 2015; Glaw & Vences, 2007; Schmidt, 1911; Schubert & Christophers, 1985; Schubert, Steffen & Christophers, 1990; Voeltzkow, 1893). Indeed, the process of collection often damages even the most intact specimens; Voeltzkow (1893) captured his specimens with bundles of cotton (‘Wattebäuschen’), and even this was not sufficient to prevent some scale loss. This, and the irregularity of Geckolepis scalation (Schmidt, 1911) makes meristics difficult to apply to them. These factors, combined with the secretive nature and cryptic colouration of this genus, have largely hindered progress on resolving its species taxonomy (Köhler et al., 2009; Lemme et al., 2013).

In their revision of this genus, Köhler et al. (2009) recognised three valid nominal species—Geckolepis maculata Peters, 1880, G. polylepis Boettger, 1893, and G. typica Grandidier, 1867—on the basis of pholidosis and morphometry, with G. petiti Angel, 1942, G. typica anomala Mocquard, 1909, and G. typica modesta Methuen & Hewitt, 1913 considered junior synonyms of G. typica, and G. humbloti Vaillant, 1887 a junior synonym of G. maculata. Four years later, Lemme et al. (2013) published a molecular phylogeny of this genus spanning many localities in Madagascar, and compared their trees with a morphological dataset. They designated 11 provisional Operational Taxonomic Units (OTUs), divided into three large clades, with a further clade represented by just one gene constituting G. polylepis, falling sister to G. typica. Their results are summarised in Fig. 1. While two clades were easily assigned to nominal species (G. typica and G. polylepis), G. maculata was difficult to place due to uncertainty surrounding its type locality (see the Supplemental Information), but they tentatively assigned it to OTU AB. The other OTUs are apparently distinct from any described species. Lemme et al. (2013) therefore estimated that Geckolepis might, once fully resolved, contain around ten species.

Figure 1 Molecular phylogeny and distribution of Geckolepis OTUs based mainly on Lemme et al. (2013).

Dotted lines in the phylogeny indicate uncertain placement. The phylogenetic position and distribution of Geckolepis humbloti is inferred from Hawlitschek et al. (2016).

Based on the DNA sequence data of Lemme et al. (2013), OTU D is the sister group of a clade containing OTU AB from northern Madagascar, and OTU C which is widespread along Madagascar’s east coast (Fig. 1). OTU AB occurs at sites north (e.g., Montagne d’Ambre) and south (e.g., Manongarivo, Nosy Be) of the Ankarana Massif, the only site where OTU D has so far been found. The genetic differentiation of OTU D is strong, amounting to 6.2% uncorrected pairwise sequence divergence in the mitochondrial 12S rRNA gene compared to OTU AB, and 7% to OTU C. Furthermore, OTU D did not share haplotypes with OTUs AB and C in the nuclear gene CMOS (Lemme et al., 2013).

Recently, Geckolepis humbloti was resurrected from synonymy with G. maculata, based on morphology, pholidosis, osteology, molecular phylogenetics, and biogeography (Hawlitschek et al., 2016). All specimens from the type locality of G. humbloti, Grand Comoro, and the other Comoro islands, are distantly related to all lineages from Madagascar except one in western Madagascar, which may belong to the ancestral population that colonised the Comoros. Thus, four species of Geckolepis are currently recognized, and since no specimens of G. humbloti were included in the molecular phylogeny of Lemme et al. (2013), the number of undescribed OTUs in this genus remains undiminished. The phylogenetic position of G. humbloti relative to the clades and OTUs of Lemme et al. (2013) is not completely clear, but it is inferred to be sister to the their Clade 3 (see Fig. 1).

Recent work has provided detailed descriptions of the external morphology of the holotypes of the four currently recognised species of Geckolepis (Hawlitschek et al., 2016; Köhler et al., 2009), and has identified genetic lineages constituting probable new species (Lemme et al., 2013). Two important taxonomic tasks remain: to firmly assign a genetic lineage to G. maculata, and to describe the outstanding cryptic lineages. The present study seeks to contribute to both of these goals, and thereby to facilitate further work on this complex. We describe a morphologically distinct form (OTU D from Lemme et al., 2013) as a new species, and provide a detailed osteological description of the genus Geckolepis based on micro-Computed Tomography (micro-CT) scans, with comparative reference to G. maculata, G. humbloti, the new species, and a member of the OTU AB from Lemme et al. (2013) to lay a foundation for osteological data as a part of the integrative systematics of this genus. We discuss the remarkably large scale size of the new species in the context of an anti-predator defence mechanism and earlier works on Geckolepis integument, and go on to highlight the next steps in the taxonomic resolution of Geckolepis.

Materials and Methods

Specimens were collected and euthanized before being fixed in 90% ethanol and transferred to 70% ethanol for long-term storage. The following institutional acronyms are used: Université d’Antananarivo Département de Biologie Animale (UADBA); Zoologische Staassammlung München (ZSM); Museum für Naturkunde, Berlin (ZMB). Field numbers (FGZC) refer to the zoological collections of F Glaw. This study involved no experiments on living animals.

Our description scheme follows the re-descriptions of Köhler et al. (2009) for direct comparability with currently recognised taxa. They make use of the following characters, which we directly replicate here (see Fig. 2): axilla to groin distance (Ax–Gr); number of canthal scales (CS) in a straight line along the canthal ridge between post-nasals and orbit; horizontal eye diameter (ED); eye-to-ear distance (EED), from posterior margin of the eye to anterior margin of the ear; head height (HH), measured at the posterior margin of the eye; head length (HL), measured from snout tip to a point level with the anterior margin of the ear opening; maximum head width (HW); number and fraction of infralabial scales (IFL), counted to one decimal place, anterior to the point level with the anterior margin of the eye; interorbital distance (IOD), measured on the dorsal surface of the head and corresponding to the narrowest point of the underlying frontal bone; number of interorbital scales at level of mid-eye (IOS); number of subdigital lamellae on free portion of the first toe (L1TF); total number of subdigital lamellae on the first toe, including the divided one adjacent to the claw (L1TT); number of conspicuously widened subdigital lamellae on the fourth toe (L4TE); total number of subdigital lamellae on the fourth toe including divided one adjacent to claw (L4TT); number of scales around midbody (MBS); horizontal length of a typical scale anterior to the eye in the loreal region (SAE); external shank length (ShL); snout length (SnL), from the tip of the snout to the anterior margin of the orbit; horizontal length of an average sized scale posterior to the eye in the temporal region (SPE); number of supralabials to the level of mideye (SPL); snout–vent length (SVL); tail length (TL); number of ventrals in one head length as defined above (VHL), counted at midventer; number of ventral scales from postmentals to vent (VS), excluding the small scales behind postmentals and those anterior to the vent. The ‘size of a typical dorsal scale at midbody (DBS)’ was not measured because ‘size’ is ambiguous; instead, we measured the width and length of a mid-dorsal scale. Additionally, the ‘number of mid-dorsal scales (DHL), from snout tip to a point level with the anterior margin of the ear opening’ was not counted because this definition is inconsistent with the values given by Köhler et al. (2009). Postmental state is given following Lemme et al. (2013), but postnasal scale states are given following Köhler et al. (2009), because they were not included in Lemme et al. (2013); see Fig. 2.

Figure 2 Schematic drawings exemplifying morphological measurements and scale counts, following the scheme of Hawlitschek et al. (2016) based on Köhler et al. (2009).

(A) Measurements and pholidosis of the head, drawn from the specimen in Fig. 3A (small scales around the eye and nostril not depicted); (B) body measurements; (C) postmental scale conditions, adapted from Lemme et al. (2013); (D) postnasal scale configuration types, adapted from Köhler et al. (2009). Scale counts in A: purple, canthal scales; blue, supralabials; red, infralabials. In C, cyan indicates mental scales and purple indicates postmental scales. For abbreviations and unillustrated characters, see Materials and Methods.

Measurements and meristics were performed by MDS using a digital calliper (0.01 mm precision) to the nearest 0.1 mm. Scale counts and finer measurements were performed using an Olympus® (Tokyo, Japan) SZX-ILLK200 stereomicroscope.

The electronic version of this article in Portable Document Format (PDF) will represent a published work according to the International Commission on Zoological Nomenclature (ICZN), and hence the new names contained in the electronic version are effectively published under that Code from the electronic edition alone. This published work and the nomenclatural acts it contains have been registered in ZooBank, the online registration system for the ICZN. The ZooBank LSIDs (Life Science Identifiers) can be resolved and the associated information viewed through any standard web browser by appending the LSID to the prefix http://zoobank.org/. The LSID for this publication is: urn:lsid:zoobank.org:pub:12C9B971-CF0F-49B7-93B1-0A480ACA9B53. The online version of this work is archived and available from the following digital repositories: PeerJ, PubMed Central, and CLOCKSS.

The following specimens were micro-CT scanned for this study: Geckolepis maculata ZMB 9655; Geckolepis megalepis sp. nov. ZSM 2126/2007 (FGZC 1144) and ZSM 289/2004 (FGZC 554); Geckolepis sp. of OTU AB sensu Lemme et al. (2013) ZSM 1520/2008 (FGZC 1697). Scans of Geckolepis humbloti produced for Hawlitschek et al. (2016) were re-analysed in this study: ZSM 81/2006 (FGZC 836) and ZSM 80/2010 (FGZC 4029). Micro-CT scans were produced using a phoenix—x nanotom® m cone-beam micro-CT scanner (GE Measurement & Control, Wunstorf, Germany) employing a tungsten or diamond target and a 0.1 mm Cu filter. Specimens were mounted and stabilised using polystyrene and small wooden braces inside polyethylene or polypropylene vessels containing a small amount of 75–80% EtOH to achieve air saturation and prevent desiccation. Scans using the tungsten target were performed at 140 kV and 80 µA; scans using the diamond target were performed at 100 kV and 100 µA. Full body scans were performed for 12 or 20 min (1,440 or 2,440 projections) with specimens mounted at an oblique angle. Skull scans were performed for 20 or 30 min with 2,440 projections at a timing of 500 or 750 ms with specimens mounted vertically. For scanning details of G. humbloti, see Hawlitschek et al. (2016). Volume renders were produced in VG Studio Max 2.2 (Visual Graphics GmbH, Heidelberg, Germany) and Avizo Lite 9.0.0 (FEI Visual Sciences Group, Burlington MA, USA). Osteological description follows terminology of recent anatomical descriptions by Daza et al. (2008), Evans (2008), Russell & Bauer (2008), and Daza & Bauer (2015) and is based on volume renders produced in VG Studio Max 2.2 and Avizo Lite 9.0.0. Skeletal figures were prepared from volume renders produced in VG Studio Max 2.2 and Avizo Lite 9.0.0. DICOM stacks of the scan files have been uploaded to MorphoSource.org and are available at http://morphosource.org/Detail/ProjectDetail/Show/project_id/302.

All field research and collecting of specimens was approved by the Malagasy Ministère de l’Environnement, des Eaux et des Forêts (Direction des Eaux et Forêts, DEF) under the following permits: 238-MINENVEF/SG/DGEF/DPB/SCBLF dated 14 November 2003; 298/06-MINENV.EF/SG/DGEF/DPB/SCBLF/RECH dated 22 December 2006; 036/08 MEEFT/SG/DGEF/DSAP/SSE dated 30 January 2008; and 174/16/MEEF/SG/DGF/DSAP/SCB, dated 25 July 2016. Export of specimens was approved by the DEF under permits: 094C-EA03/MG04, dated 1 March 2004; 051N-EA03/MG07, dated 10 March 2007, and 270N-EA09/MG16, dated 7 September 2016.

Results

Our analysis of three specimens of Geckolepis from the Ankarana Reserve assigned to OTU D by Lemme et al. (2013) confirmed that these individuals are distinct in pholidosis from any described species, having a lower number of larger scales than any other known populations of Geckolepis. Micro-CT scans of the skeletons of these individuals and several congeners reveal subtle differences in osteology between known species and lineages. Consequently, we here describe OTU D as a new species on the basis of morphometrics, pholidosis, skeletal morphology, and molecular phylogenetics, and compare its skeletal morphology in detail with those of individuals of G. maculata, G. humbloti, and OTU AB sensu Lemme et al. (2013).

Geckolepis megalepis sp. nov.	
Geckolepis sp. OTU D—(Lemme et al., 2013)	
(Figs. 3–9, Table 1, Appendices S1 and S2, Video S1, Fig. S1)	
LSID: urn:lsid:zoobank.org:act:08097CA5-172E-4F45-AE68-AC4E02BBC644	

Holotype. ZSM 2126/2007 (FGZC 1144), an adult of unknown sex, from the east side of Ankarana National Park (12.9564°S, 49.1172°E, ca. 150 m a.s.l.), Antsiranana Province, north Madagascar, collected on 3 March 2007 by P Bora, H Enting, F Glaw, A Knoll & J Köhler.

Figure 3 Specimens of Geckolepis megalepis sp. nov. in life.

(A) A specimen observed by A. Anker (photograph used with permission); (B) a specimen observed by FG, and (C) a specimen photographed after scale loss, with inset indicating the transparent ‘tear zone’ at the base of a scale. None of the photographed animals were collected, but their attribution to G. megalepis is clear on the basis of the large size of their scales. Note that the tails of all three specimens are regenerated.

Figure 4 Specimens of Geckolepis megalepis sp. nov. in preservative.

(A–C) The holotype, ZSM 2126/2007; (D–F) paratype ZSM 232/2016. Shown in dorsal (A, D) and ventral (C, F) view, with a close-up view of the postmental scales (B, E) coloured for reference to Fig. 1. Scale bars indicate 10 mm. Chin insets are not to scale.

Figure 5 Micro-CT images of mineralized scales of Geckolepis specimens.

Shown in dorsal (A, D, F, H, J) and ventral (B, E, G, I, K) view, and coronal cut at the parietal level (C) of Geckolepis megalepis (A–C, ZSM 2126/2007), Geckolepis OTU AB (D–E, ZSM 1520/2008), G. maculata (F–G, ZMB 9655), and G. humbloti (H–I, ZSM 80/2010; J–K, ZSM 81/2006). Note the high density of the scale covering in all the species, and the lack of mineralization in the postmental scales (compare with Figs. 2 and 3).

Paratypes. ZSM 289/2004 (FGZC 554), probably a subadult, sex unknown, from between Mahamasina and the Petit Tsingy (exact coordinates not known, but ca. 12.9558°S, 49.1181°E, ca. 125 m a.s.l.), Ankarana National Park, Antsiranana Province, north Madagascar, collected 25 February 2004 by F Glaw, M Puente & R Randrianiaina; ZSM 232/2016 (FGZC 5476), an adult of unknown sex, from the private forest of the Ankarana Lodge (12.9613°S, 49.1499°E, 134 m a.s.l.), Ankarana massif, Antsiranana Province, north Madagascar, collected 28 August 2016 by F Glaw, K Glaw, T Glaw, Jaques & NA Raharinoro; FGZC 1606 (UADBA uncatalogued), sex and age unknown, from Petit Tsingy (ca. 12.9558°S, 49.1181°E, ca. 125 m a.s.l.), Ankarana National Park, Antsiranana Province, north Madagascar, collected 12 February 2008 by N D’Cruze, M Franzen, F Glaw & J Köhler.

Table 1 Comparative morphological data on Geckolepis megalepis, and the holotype of G. maculata (ZMB 9655), with values for G. polylepis and G. typica given by Köhler et al. (2009).

For raw measurements of individual G. megalepis specimens, see Appendix S1.

Species:	G. megalepis sp. nov.	G. maculata	G. polylepis	G. typica	
No. Specimens	3	Only ZMB 9655	11	102	
SVL (mm)	57.8–69.5 (65.2 ± 6.46)	58.5	41.1–50.2 (46.5 ± 2.8)	31.1–56.3 (43.0 ± 5.02)	
TL (mm)	71.7–80.1 (75.3 ± 4.33)	N/A	34.5–56.5 (48.0 ± 6.4)	29.7–58.5 (43.6 ± 6.83)	
Ax–Gr (mm)	23.9–32.5 (27.9 ± 4.34)	23.7	14.7–24.3 (19.8 ± 2.7)	12.4–28.3 (19.0 ± 2.76)	
ShL (mm)	5.0–7.0 (5.9 ± 1.01)	6.2	5.5–7.2 (6.5 ± 0.5)	4.0–8.2 (5.7 ± 0.72)	
HL (mm)	14.9–17.1 (16.0 ± 1.10)	15.2	9.7–12.0 (10.7 ± 0.8)	6.9–12.8 (9.6 ± 1.08)	
HW (mm)	13.3–18.5 (16.0 ± 2.60)	14.2	9.1–12.0 (10.6 ± 0.9)	6.4–14.2 (9.8 ± 1.43)	
HH (mm)	7.4–10.1 (8.80 ± 1.35)	7.6	4.8–6.4 (5.8 ± 0.5)	3.4–8.8 (5.1 ± 0.77)	
SnL (mm)	7.0–7.9 (7.5 ± 0.47)	7.0	4.8–5.8 (5.1 ± 0.3)	3.2–6.0 (4.4 ± 0.54)	
ED (mm)	4.3–4.3 (4.3 ± 0.00)	3.7	2.2–3.3 (2.6 ± 0.3)	1.8–3.2 (2.4 ± 0.29)	
IOD (mm)	7.7–9.6 (8.7 ± 0.95)	7.0	4.6–6.4 (5.3 ± 0.5)	3.5–6.7 (4.9 ± 0.64)	
EED (mm)	4.9–6.3 (5.4 ± 0.78)	4.8	2.8–4.0 (3.4 ± 0.4)	2.2–4.1 (3.1 ± 0.41)	
IOS	9–10 (9.67 ± 0.58)	9	9–12 (10.8 ± 1.09)	9–13 (10.6 ± 0.93)	
SPL	7–8 (7.33 ± 0.52)	6	6.0–7.0 (6.91 ± 0.30)	5.0–7.0 (6.06 ± 0.44)	
IFL	4.5–4.8 (4.6 ± 0.15)a	4.1	4.0–5.5 (4.65 ± 0.45)	2.8–4.6 (3.72 ± 0.39)	
CS	6–7 (6.7 ± 0.58)	6	6–7 (6.1 ± 0.30)	4–7 (5.5 ± 0.58)	
MBS	17–18 (17.7 ± 0.58)	25	30–37 (34.5 ± 2.46)	26–36 (30.9 ± 1.98)	
VS	27–31 (29.0 ± 2.00)	32	40–50 (44.6 ± 3.01)	37–48 (41.6 ± 0.22)	
VHL	6.0–8.0 (7.0 ± 1.00)	9	7.0–13.0 (10.41 ± 1.76)	7.0–12.0 (8.78 ± 1.11)	
L1TT	14–15 (14.33 ± 0.58)	12	8–15 (10.4 ± 1.86)	8–13 (9.6 ± 1.03)	
L4TT	18–21 (19.67 ± 1.53)	17	10–18 (15.3 ± 2.31)	11–18 (14.9 ± 1.09)	
SAE/SPE	0.42–0.48 (0.45 ± 0.03)	0.49	0.67–1.00 (0.79 ± 0.01)	0.50–0.80 (0.63 ± 0.07)	
Notes.

Ranges for G. polylepis are probably reliable, but those for G. typica certainly come from several OTUs, and are therefore not actually representative of the variation of that species. Values for true G. typica, as for G. maculata, are currently certain only from its holotype.

aThis value includes the specimen figured in Figs. 2 and 3A, and has a sample size of 4.

Diagnosis. A species of the genus Geckolepis based on its overall morphology and large, fish-like scales (similar to cycloid scales in terms of the extent of overlap), as well as its phylogenetic position (Lemme et al., 2013; Fig. 1). Geckolepis megalepis differs from all of its congeners by the possession of the following suite of characters: innermost pair of postmental scales in broad contact (condition A/B, Fig. 4), SVL ≤ 69.5 mm, infralabials to anterior margin of eye 4.5–4.8, 17–18 scales rows around the midbody, 27–31 ventral scales between the postmentals and the vent, and the absence of a dark lateral stripe, and typical midbody dorsal scales measuring 7.3–8.3% of the SVL in length. Osteologically, G. megalepis is characterised by a narrow infraorbital fenestra, a bulging nasal cavity, nasals with straight sides, a well developed anterior extension of the subfrontal process, a notched premaxilla-vomer fenestra, scapular ray of scapulocoracoid not surpassing the clavicle, and posteriorly curved pubic tubercle of the pubis. Additionally, it is separated by an uncorrected pairwise genetic distance in the mitochondrial ND4 gene of ≥10.1% from all other lineages of Geckolepis and has a unique CMOS haplotype (Lemme et al., 2013).

Geckolepis megalepis may be distinguished from G. maculata (note: because of the substantial uncertainty surrounding the identity of G. maculata, we here compare G. megalepis only to the holotype of that species, ZMB 9655, until such a time as its true affinities can be clarified; see Köhler et al. (2009) for a detailed morphological account of that specimen) by the combination of fewer scale rows around midbody (17–18 vs. 25), fewer ventral scales (27–31 vs. 32), larger relative scale size (typical midbody dorsal scale 7.3–8.3% of SVL vs. 5.4%), and the absence of a dark lateral head stripe (vs. presence); from G. typica by larger maximum size (SVL up to 69.5 mm vs. <57 mm), fewer scale rows around midbody (17–18 vs. 28–32), fewer ventral scales (27–31 vs. 42–49), postmental scale condition (A/B vs. D), and the absence of dark longitudinal stripes on the dorsum (vs. presence); from G. polylepis by larger maximum size (SVL up to 69.5 mm vs. <52 mm), fewer scale rows at midbody (17–18 vs. 30–37), fewer ventral scales (27–31 vs. 37–55), and the absence of dark longitudinal stripes on the dorsum (vs. presence); and from G. humbloti by fewer scale rows around midbody (17–18 vs. 22–30), more infralabials to the anterior margin of the eye (4.5–4.8 vs. 3–4), and fewer ventral scales (27–31 vs. 33–41).

For comparison of the osteology of the new species with Geckolepis maculata, G. humbloti, and a specimen of OTU AB, see the Osteology of Geckolepis section below.

Description of the holotype. (Fig. 4) A large specimen in a moderately good state of preservation. Several scales missing from dorsum and the left knee, and a few older scars on the venter; tail detached but preserved, presumably autotomized during or after capture.

SVL 68.4 mm; tail length 80.1 mm; axilla-groin distance 27.2 mm; shank length 5.8 mm; head length 15.9 mm; head width 16.1 mm; head height 8.9 mm; snout length 7.7 mm; eye diameter 4.3 mm; interorbital distance 8.7 mm; eye-ear distance 5.0 mm; rostral large, convex, distinctly visible from above; large postrostrals (4 supranasals) separated by a thin oblong scale anteriorly and two small scales posteriorly (type D); nostril bordered by rostral, first supralabial, three postnasals, and one supranasal (=postrostral); postnasals approximately equal in size to anterior loreals; postnasals and loreals separated by one row of small scales; scales on snout and on loreal region almost all with triangular posterior margins, slightly convex, imbricate; seven canthal scales in a line on the canthal ridge, including the postnasals, between nostril and anterior margin of eye; scales at supraorbital region similar to and continuous with those on top of head; scales increase by a factor of 1.68 from level of mideye to occipital region; ten scales along a straight line dorsally between orbits; three rows of small scales adjacent to anterior margin of eye, decreasing to two on the upper and two on the posterior margin of eye; pupil vertical; seven supralabials to below centre of eye (eight total) on left side, eight to below centre of eye (nine total) on right side, all of roughly equal size, the posterior-most two smaller than the rest; scales in temporal region larger by factor 2.25 than those in loreal region; ear opening much smaller than eye, horizontally oval; 4.5 infralabials to below level of anterior margin of eye, decreasing in size posteriorly; mental scale large, triangular, with convex anterior border; postmentals asymmetric: one large pair immediately posterior to mental scale with broad medial contact, followed on the right side by two postmentals of decreasing size, and on the left by one broader postmental with an irregular posteromedial border (condition A/B; Fig. 4); one row of small scales separating postmentals from anterior chin scales; chin and ventral body scales rhomboid with rounded triangular posterior margins, imbricate, gradually increasing to double size posteriorly, arranged in roughly regular rows; 29 scales along the midventral line (count somewhat inhibited by ventral scarring) between postmentals and vent, not including smaller scales adjacent to postmentals and anterior to vent; dorsal scales cycloid, larger than lateral or ventral scales; 18 scales around midbody; anterior half of tail flat and rather wide (55% of body width), decreasing in width gradually, down to a thin tip; tail covered with scales similar to body scales, but gradually decreasing in size posteriorly, a series of transversely expanded median subcaudals present, starting roughly 20 mm from the base of the tail; 14 lamellae under first toe, extending from the sole of the foot to the claw, and 21 under fourth toe, 12 of which are noticeably expanded; claws exposed, non-retractile. A single midbody scale measures 5.0 mm wide by 5.8 mm long, and is therefore 7.8% of the SVL in length.

Colouration in preservative: Head dorsally homogenously greyish brown, laterally greyishbrown flecked with darker and lighter areas posterior to eye and below mouth; no obvious dark lateral stripe; dorsal scales greyish brown flecked with dark and pale spots; legs as dorsum; ventrally dirty white; tail greyish brown with four dark transverse markings that do not continue onto the whitish ventral surface; exposed dorsal skin brown, ventral skin whitish. No information exists regarding the life colouration of the specimen.

Variation. (Figs. 3 and 4) Both ZSM paratypes strongly resemble the holotype, but one (ZSM 289/2004) is much smaller and presumed subadult. Their measurements and meristics are provided as part of Table 1, and in full detail in Appendix S1. These specimens differ from the holotype in the following characters not provided in Appendix S1: scales increase by a factor of 2.12–2.13 from level of mideye to occipital region; eight or nine total supralabials on each side; postnasal configuration of ZSM 232/2016 type A, 289/2004 type D; postmentals large, three or four pairs present: one large pair immediately posterior to mental scale with broad medial contact, followed two or three pairs of decreasing size (condition A); mid-dorsal scales range from 7.3% to 8.3% of SVL; tail of ZSM 289/2004 not especially broadened (regenerated) and greyish brown with three darker transverse markings that do not continue onto the whitish ventral surface; tail of ZSM 232/2016 broad at its base and narrows rapidly (at least partially regenerated); overall body colouration of ZSM 232/2016 is greyer than the other specimens, though this may be because it was collected more recently. In ZSM 289/2004, one scale on the dorsal surface of the neck and one on left side of dorsal tail base are dark brown with a burned appearance. The animal in Fig. 3A has 4.8 infralabials to the anterior edge of the eye (see Fig. 2A), and we thus infer that this value can range from 4.5 to 4.8. From Figs. 3A and 3B, the colouration in life is assessed to be mostly grey with dark spots on some scales, giving a mottled appearance.

Phylogenetic relationships. Geckolepis megalepis is closely related to a sister species pair formed by OTUs C and AB (Lemme et al., 2013; Fig. 1), which are widely distributed in eastern and northern Madagascar. Their taxonomic status will need to be assessed in more detail in future work on this genus.

Habitat, natural history, and conservation status. Geckolepis megalepis was observed active at night both in the rainy and dry seasons, on trees (see Figs. 3A–3B) and tsingy limestone rock. When captured, these geckos showed a strong tendency to autotomize large parts of their scales, leading to partly ‘naked’ geckos without any visible (bloody) lesions (Fig. 3C). In a subjective comparison this tendency appeared to be even more developed than in other Geckolepis species.

The new species occurs syntopically in Ankarana with at least one additional Geckolepis species (OTU G). It is only known from the dry deciduous forest among the limestone tsingy karst of Ankarana Reserve and its immediate vicinity, an area of 182 km2. Due to its likely limited distribution (182 km2), knowledge from only two threat-defined localities in the Ankarana massif, and the potential for rapid decrease in quality of the forests of that reserve and the area around it due to illicit deforestation, anthropogenic fire, sapphire mining, and free-ranging grazing of livestock (e.g., Hawkins et al., 1990) we propose that it be listed as Near Threatened under the IUCN criteria (IUCN, 2012).

Two other geckos endemic to Ankarana Reserve are assessed as Near Threatened (Lygodactylus expectatus Pasteur & Blanc, 1967) and Endangered (Phelsuma roesleri Glaw et al., 2010). We defend the choice of Near Threatened instead of Endangered for G. megalepis on the following grounds: although it satisfies IUCN criterion B1 sub-criterion a, it fails to qualify for Endangered under sub-criteria b or c, as we can only identify potential threats; were these to be realised, then the species would immediately qualify for Endangered, but until that point, it remains Near Threatened. The same cannot be said of P. roesleri, as it lives on Pandanus plants, which are less common in Ankarana and potentially harvested by humans, making the risk to it greater, while G. megalepis is more generalist in its habits. Our recent observation of several individuals in a short timespan suggests that the population of Geckolepis megalepis in Ankarana is at least locally healthy.

Etymology. The specific epithet is derived from the two Greek stems με ´γας (mégas) meaning ‘very large’ and λεπι ´ς (lepís) meaning ‘scale’, and refers to the large size of the scales of this species in comparison to its congeners and other geckos, which aids also in its diagnosis.

Available names. Three junior synonyms currently exist within the genus Geckolepis that must be considered as possible earlier names for G. megalepis: G. typica anomala Mocquard, 1909, G. typica modesta Methuen & Hewitt, 1913, and G. petiti Angel, 1942. Synonymy of G. anomala, G. modesta, and G. petiti with G. typica was discussed at length by Köhler et al. (2009). While this placement needs to be re-analysed in light of the genetic information produced by Lemme et al. (2013), Geckolepis megalepis can be distinguished easily from the type series of G. anomala, G. modesta, and G. petiti by its postmental scales (condition A/B vs. D in G. anomala, G. modesta, and G. petiti), and fewer scale rows at midbody (17–18 vs. 32 in G. anomala, 22–25 in G. modesta, and 28 in G. petiti).

Remarks. The living specimen depicted in Fig. 3 of Lemme et al. (2013) as OTU D is misattributed and does not belong to Geckolepis megalepis. Lemme et al. (2013) report 17–20 scale rows at midbody for this species; the reason for this discrepancy could not be established here, but we are confident in our counts. However, we also emphasise that their higher number would remain diagnostic in all of the comparisons presented above. Köhler et al. (2009) probably did not include any specimens of Geckolepis megalepis in their revision of the genus, as they did not consider any individuals with fewer than 22 scale rows at midbody.

Osteology of Geckolepis

Osteological comparisons. The scales of Geckolepis geckos are mineralized and resemble osteoderms (Fig. 5; see also Schmidt, 1911). Among gekkotans, only Gekko gecko and Tarentola species (Bauer & Russell, 1989; Daza & Bauer, 2015; Schmidt, 1911; Vickaryous, Meldrum & Russell, 2015) are known to develop similarly mineralised integumentary coverings. The dense scales of Geckolepis differ from the osteoderms of G. gecko and T. mauritanica in that they are imbricate, and not juxtaposed and adpressed against the skull bones. Geckolepis also differ from the majority of extant gekkotans in having paired and unfused (both dorsally and ventrally) frontal bones; we only found fused frontal bones in a large specimen of the OTU AB sensu Lemme et al. (2013) from Montagne des Français (‘AB specimen’ henceforth), which is the most osteologically distinct specimen from our sample.

Geckolepis megalepis and the AB specimen differ from other Geckolepis in having a narrow infraorbital fenestra. In these two taxa, the nasal cavity also bulges slightly more than in smaller Geckolepis specimens. There is some variation in the shape of the nasal bones, being rectangular (with straight sides) in G. megalepis and the AB specimen. Geckolepis humbloti has nasal bones with a sigmoid lateral edge instead of straight. The holotype of G. maculata has nasals with straight lateral edges (Fig. 10). Geckolepis megalepis and the AB specimen have a more anterior extension of the subfrontal process of the frontal in palatal view, fused in the AB specimen and not fused in G. megalepis; all others have a large notched area that does not extend anteriorly. Another distinct feature of G. megalepis was found in the shape of the premaxilla-vomer fenestra, being notched instead of rounded as in other Geckolepis.

Figure 6 Micro-CT images of skull (cranium and jaw) of the holotype of Geckolepis megalepis sp. nov. (ZSM 2126/2007).

(A) Dorsal, (B) ventral, (C) lateral, (D) labial, and (E) lingual view. Abbreviations: adf, anterior inferior dental foramen; amf, anterior mylohyoid foramen; ar, articular facet; asnp, ascending nasal process of premaxilla; bo, basioccipital; bp, basipterygoid process of parabasisphenoid; ch, choana; cob, compound bone; cor, coronoid; d, dentary; dpp, descending parietal process; ect, ectopterygoid; ept, epipterygoid; f, frontal; hscc, horizontal semi-circular canal; if, incisive foramen; j, jugal; larst, lateral aperture of the recessus scalae tympani; mdf, mandibular fossa; mf, mental foramen; mko, opening of the Meckelian canal, mx, maxilla; mx.ap, anterior process of maxilla; mx.fp, facial process of maxilla; mx.pp, posterior process of maxilla; mx.ps, palatal shelf of maxilla; n, nasal; occ, occipital condyle; oto, otooccipital; pal, palatine; pal.mp, maxillary process of palatine; pal.vf, vomerine flange of palatine; par, parietal; pbsh, parabasisphenoid; pmx, premaxilla; pmx.pps, premaxilla palatal shelf; pof, postorbitofrontal; pop, paroccipital process; ppp, posterolateral process of parietal; prf, prefrontal; pro, prootic; pro.ca, crista alaris of prootic; pro.cp, crista prootica of prootic; psaf, posterior surangular foramen; pscc, posterior semicircular canal; pt, pterygoid; pt.qp, quadrate process of pterygoid; q, quadrate; q.ch, conch of quadrate; rap, retroarticular process; s, stapes; saf, surangular foramen; scr, sclerotic ring; smx, septomaxilla; spl, splenial; sof, suborbital fenestra; sq, squamosal, so, supraoccipital; spht, sphenooccipital tubercle; tbr, trabeculae; v, vomer; v.lp, lateral process of vomer; v.mp, maxillary process of vomer. A rotational video of the skull is provided in Video S1.

Figure 7 Micro-CT images of spinal column of Geckolepis, based on G. megalepis sp. nov. (ZSM 2126/2007).

(A) Lateral and (B) dorsal view. Abbreviations: lb, lumbar; sc, sacral. Regions are indicated by the posterodorsal-most point of the neural arches.

Figure 8 Micro-CT images of pectoral girdle of Geckolepis, based on G. megalepis sp. nov. (ZSM 2126/2007).

(A) Ventral and (B) lateral view. Abbreviations: 1cf, primary coracoid fenestra; 2cf, secondary coracoid fenestra; acr, acromion process; cf, clavicular fenestra; cl, clavicle; cor, coracoid; ic, interclavicle; gf, glenoid fossa; hu, humerus; hu.c, humeral condyle; hu.dtp, deltopectoral crest of humerus; hu.ec, ectepicondylar crest of humerus; hu.hc, humeral crest of humerus; mc, metacarpal; ms, mesosternum; p, phalanges; ps, presternum; pu, patella ulnaris; r, radius; rl, radiale; s, scapula; sc.1c, primary coracoid ray of scapulocoracoid; sc.2c, secondary coracoid ray of scapulocoracoid; scf, scapulocoracoid fenestra; sf, scapular fenestra; ss, suprascapula; sc.sr, scapular ray of scapulocoracoid; sr, sternal ribs; sucf, supracoracoid foramen; u, ulna; ul, ulnare; xs, xiphisternum.

Figure 9 Micro-CT images of pelvic girdle of Geckolepis, based on G. megalepis sp. nov. (ZSM 2126/2007).

(A) Dorsal and (B) lateral view. Abbreviations: ac, acetabulum; ag, astragalocalcaneum; ep, epipubis; f, femur; fb, fibula; f.it, internal trochanter; f.itf, intertrochanteric fossa; f.fc, femoral condyle; hyi, hypoischium; il, ilium; il.pap, preacetabular process of ilium; is, ischium; is.t, ischiadic tuberosity; ltp, lenticular tibial patella; mt, metatarsals; of, obturator foramen; p, phalanges; pic, proischiadic cartilage; pb, pubis; pb.pl, processus lateralis of pubis; pc, post-cloacal bone; pt, pubic tubercle; tb, tibia; tf, thyroid fenestra.

Figure 10 Comparative micro-CT images of the skull (cranium and jaw) of other Geckolepis species.

Shown in dorsal (A, F, K, P), ventral (B, G, L, Q), lateral (C, H, M, R), labial (D, I, N, S), and lingual (E, J, O, T) view. Depicting Geckolepis OTU AB (A–E, ZSM 1520/2008), holotype of G. maculata (F–J, ZMB 9655), and G. humbloti (K–O, ZSM 80/2010; P–T, ZSM 81/2006). From volume-rendering of micro-CT scans. Rotational videos of these skulls are provided in Videos S2–S5. For labels, see Fig. 6.

There are also some differences between G. megalepis and the other species examined in the postcranium: the lateral processes of the first five caudal vertebrae (pygial series) are curved laterally (vs. straight in G. maculata and G. humbloti); the scapular ray of the scapulocoracoid does not surpass the clavicle (vs. surpassing the clavicle in G. maculata); the secondary coracoid ray of the scapulocoracoid extends to the level of the posterior margin of the clavicular fenestra (vs. surpassing the posterior margin of the fenestra in G. maculata); and the pubic tubercle of the pubis is posteriorly curved (vs. more or less vertical in G. maculata and G. humbloti).

For future reference we listed 24 variable skull traits (Appendix S2) among Geckolepis that provide a baseline for future comparative studies.

Skeletal description. In the following section, we present a generalised skeletal description of the genus Geckolepis. Data on G. humbloti is based on the scans produced for Hawlitschek et al. (2016), re-analysed for this study. The postcranial skeleton of the AB specimen was not assessed; our postcranial osteological description pertains only to G. humbloti, G. maculata, and G. megalepis.

As we have mentioned above, one key feature of Geckolepis is the presence of a dense covering of mineralized scales, (Fig. 5). These mineralized scales contrary to the osteoderms of Gekko gecko and Tarentola mauritanica (Vickaryous, Meldrum & Russell, 2015) are imbricate and not adpressed against the skull. Schmidt (1911) referred to these scales as osteoderm, but noted that they are unique in lacking bone cells (that is to say, they are not osteoderm in the strict sense). He found that these mineralizations, which he showed to be formed from calcified tissue fibres, roughly trace the outline of the scales, but are mostly confined to their middles and do not extend into the keratinous scale. Our micro-CT data indicates that mineralization of scales is more extensive, at least in the examined species; scales shown in Fig. 5 are very similar in dimensions to what they look like in life (e.g., compare Fig. 5A with ZSM 2126/2007 in Fig. 4). The reasons for this discrepancy are not clear, and will require further study. However, we can confirm that these mineralizations do not extend to the tip of the scales, as can be seen by the soft-looking distal edges of the scales in Fig. 5. Schmidt (1911) also found that mineralization was lacking from specific scales, including the labials and postmentals and scales proximal to these, small scales of the head including those around the eye and ears, and fingers and toes; they also decrease in frequency in the tail scales beyond the first third. This pattern is recapitulated by our micro-CT scans. In a new study by Paluh and colleagues (in review), these mineralized scales are again suggested to be osteoderms, but we conservatively refer to them here as ‘mineralized scales’ until their study is published.

The mineralized scales were digitally removed from all the CT scans to facilitate rendering of the underlying bone surface and sculpturing. We also digitally removed the endolymphatic sacs.

Skull (Figs. 5–10): The skull of Geckolepis is the typical broad and depressed skull of geckos (Kluge, 1967), wedge shaped in lateral view. The left maxilla of the holotype of G. maculata (ZMB 9655) is fractured, and the premaxillary and maxillary palatal shelves show some irregular holes. There is no trace of fracture bones associated with these holes, so the cracks may have been caused by an infection that healed during the animal’s life, or may have resulted from fixation in formalin and gradual decalcification over almost 150 years (although the skull and postcranial skeleton shows no other signs of extensive decalcification). The remaining specimens examined are intact (G. megalepis, G. humbloti, and the AB specimen). Figure 6 provides anatomical labels for most features based on the holotype of G. megalepis, whereas Fig. 10 provides comparative images of one adult specimen each of the three other species included for comparative purposes. Rotational videos of these scans are provided as Videos S1–S5. The skeleton of G. megalepis paratype ZSM 289/2004 is not figured, as the resolution of our micro-CT scan of it is too low, and only some of its character states could be accurately determined (see Appendix S2); these however largely agree with the holotype.

Cranium: Nares oriented anteriorly, bordered medially by premaxilla, ventrally by premaxilla and maxilla, laterally by facial process of the maxilla, and dorsally by nasals. The orbits are incomplete posteriorly, and they accommodate the majority of the circular eye (as defined by the sclerotic ring). The orbits are oriented anterolaterally, possibly enabling some field of vision overlap. They are formed by the maxilla and jugal ventrally, prefrontal anteriorly, frontal dorsally, and postfrontal posterodorsally. A sclerotic ring is present, composed of 14 bones.

The premaxilla is fused, with isodont, sharply pointed teeth with 13 tooth loci, this being a constant number among all Geckolepis specimens examined. The ascending nasal process is short and forms a bony septum between the nares, tapering abruptly dorsally, where it briefly overlaps the nasals. The palatal shelf contacts the vomer, defines an incisive foramen, and contacts the maxillae laterally.

The maxilla possesses a large facial process and a relatively narrow palatal shelf, as well as a long posterior process, an anterior process, and an anterior maxillary lappet on the lingual side of the anterior process. The alveolar border bears deeply pleurodont, sharply pointed isodont teeth. Tooth loci fluctuate between 35 and 40, 36 in G. megalepis and the AB specimen; G. maculata presents the lowest tooth count, with 35 tooth loci. The maxilla is pierced by four to six supralabial foramina. The posterior process is in contact with the jugal and ectopterygoid posteromedially. The palatal shelf contacts the anterior lateral process of the palatine posteriorly. The maxillary lappet contacts the vomer laterally and the premaxilla’s posterior palatine shelf ventrally, and does not extend to meet its contralateral. The anterior process contacts the premaxilla. The facial process is broad and dorsolaterally convex, its dorsal margin sloped downward from its posterior end to its anterior end, its posterior margin weakly (G. maculata, one specimen of G. humbloti) or strongly curved (G. megalepis, the AB specimen, and one specimen of G. humbloti) and in contact with the prefrontal; posterodorsally in contact with the frontal, and dorsally in contact with the nasal.

The nasal is nearly rectangular (except by the curved anterior edge that forms the posterodorsal margin of the nares), a small portion of the medial edge lies beneath the ascending nasal process of the premaxilla, and the anterolateral margin borders a small gap with the facial process of the maxilla; the lateral edge is straight in G. maculata, bulges slightly outward in G. megalepis and the AB specimen, and is curved with a lateral flange overlapping the maxillary facial process in G. humbloti (as seen in other geckos; Evans, 2008); laterally in broad contact with maxillary facial process, and posteriorly in contact with the frontal. Nasals are partially fused in the AB specimen.

The prefrontal is strongly convex and has an extensive overlap with the facial process of the maxilla, leaving the exposed surface roughly crescent-shaped in all species (slightly more crescentic in G. megalepis and some individuals of G. humbloti). The posterior edge is weakly bowed and curves posteromedially forming the orbito-nasal flange. Dorsally it is distantly separated from the postorbitofrontal. The prefrontal and the maxilla bound the lacrimal foramen.

The frontals remain paired and unfused—this state may however change with age, as the AB specimen has at least partially fused frontals, although a partial suture is still visible anteriorly and posteriorly (see Fig. 10A). It is in anterior contact with the nasal (straight or slightly concave suture), lateral contact to the facial process of the maxilla (concave suture) and the prefrontal (convex suture), posterolateral contact with the postorbitofrontal (which clasps the frontoparietal suture), and an extensive frontoparietal suture that is weakly curved anteriorly. The anterior end is overlapped by the nasal bones, and the visible portion is roughly half the width of the posterior end, the narrowest point being at the interorbital point. The subolfactory processes of the frontals contact each other but they remain separated, so there is also no ventral fusion. This condition is extremely rare in gekkotans, known only in the fossil Gobekko cretacicus (Daza, Bauer & Snively, 2013). The crista cranii of the frontals are sutured to the medial side of the posterodorsal process of the prefrontal, thereby forming the dorsal and anterior orbital ridge. The frontal lacks significantly extended anteromedial and anterolateral processes.

The jugal is elongated and slender with tapered ends. It extends from the posterior process of the maxilla anteriorly along its medial edge, in contact with the ectopterygoid ventromedially, almost extending far enough forward to meet the palatine and participate in the lacrimal foramen.

The parietal is in broad medial contact with its contralateral, the suture is straight in G. maculata and zigzags in G. megalepis, G. humbloti, and the AB specimen, although in the lattermost there is also partial fusion, rendering the suture faint. The parietal also contacts the postorbitofrontal anterolaterally, crista alaris of the prootic lateroventrally, and the squamosal posterolaterally. The bone is broad, curved downwards forming some lateral protection for the brain. It is subtrapezoidal in shape, its lateral and median margins subparallel, the anterior margin angled posteriorly along the frontoparietal suture, the posterior margin angled anteriorly. The posteroparietal process is long and thin in G. maculata and one specimen of G. humbloti, and broad and short in G. megalepis, one specimen of G. humbloti and the AB specimen, extending posterolaterally from the posterolateral corner of the parietal to contact with the squamosal.

The postorbitofrontal is laminar: thin, short, and curved, extending just anterior and posterior to frontoparietal suture and bracing it (Daza et al., 2008; Rieppel, 1984) in contact with frontal anteromedially and parietal posteromedially. It lacks a discrete free process for the attachment of the postorbital ligament (Evans, 2008), which might instead be anchored to the body of the bone.

The squamosal is short, slender, and curved, contacting the posterolateral process of the parietal anteromedially, and the paroccipital process posteriorly. It is considerably reduced in G. maculata.

The quadrate has a deep indentation in the conch. This bone meets the quadrate process of the pterygoid ventrally and has suspension formed by ligaments of the squamosal and the paroccipital processes; it is not in direct contact with any other bones. It has a thick central column and a thin, posterolaterally directed conch that lacks an obvious squamosal notch dorsally. Its cephalic condyle is dorsomedial and not strongly expanded. Its mandibular condyle is concave. It possesses a large foramen in the ventral half of the conch.

The septomaxilla is very thin, U-shaped, in anterior contact with the premaxilla, otherwise suspended in the nasal capsule. Its medial arm contacts the contralateral, separated anteriorly by a small foramen lying dorsal to the incisive foramen of the vomer. The lateral arm ascends slightly, and is long and cuneate with a sculpted lateral surface.

The vomer is a thin, laminar bone. It is strongly fused to its contralateral, though a median ridge allows the individual bones to be distinguished despite strong ontogenetic fusion. Anteromedially an incisive foramen is present between the vomer and the posterior palatal shelf of the premaxilla, the shape of this foramen is variable among the species examined, being a v-shaped notch (G. megalepis and G. maculata) or oval shaped (G. humbloti and the AB specimen). The anterolateral extension of the vomer is in contact with the lingual maxillary lappet. Posteriorly it bears two slender lateral processes, and the paired elements form a broad median projection. The lateral process curves medially to join the vomerine process of the palatine. The median spur is bordered on either side by the distal tips of the palatine vomerine processes, and forms the anterior end of the interpterygoid vacuity. The vomer has also two foramina that might correspond to openings of the lacrimal duct.

The palatine is squarish, with rounded lateral and medial edges. The vomerine flange and maxillary process are slender and subequal in length, together forming the border of the choana. The vomerine flange lies parallel to the posterior processes of the vomer and rests on a notch on the body of the vomer; the maxillary process contacts the maxilla’s palatal shelf laterally. The palatine forms the anteromedial border of the suborbital fenestra. The bone is without an obvious pterygoid process but possesses a posteroventral shelf where the palatine process of the pterygoid overlaps it. Lateral to this overlap, the bone borders a slit extending medially from the suborbital fenestra between the palatine and pterygoid. The medial edge of the palatine forms the lateral border of the interpterygoid vacuity. The lateral face of the palatine is in contact with the anterolingual end of the ectopterygoid. The pterygopalatine joint is oblique.

The pterygoid is roughly y-shaped, with a brief anteromedial articulation with the palatine and anterodorsal articulation with the ectopterygoid, articulating with the epipterygoid at the fossa columellae, and contacting the quadrate posterolaterally. The pterygoid has a palatine process anteromedially and a sculpted anterior border that is straight lateral to the palatine process, forming the posterior border of the slit extending medially from the suborbital fenestra—then concave, forming the posterior border of the suborbital fenestra—then extending anteriorly again to form the pterygoid flange in contact with ectopterygoid, practically excluding the ectopterygoid from the posterior margin of the suborbital fenestra; also forming the posterolateral border of the interpterygoid vacuity. The facet that contacts the basipterygoid process is porous. In lateral view, the quadrate process curves laterally beyond this point and the fossa columellae to below the quadrate.

The ectopterygoid is bent downward. It is anterolaterally in contact with the jugal, and posteriorly in ventrolateral contact with the anterolateral pterygoid flange. The bone’s downward bend is due to the more dorsal position of palatine and maxilla relative to the pterygoid. Its medial margin is sigmoid in G. maculata and G. humbloti, variable in G. megalepis (wavy in ZSM 2126/2007 but sigmoid in ZSM 289/2004), and wavy in the AB specimen. The suborbital fenestra is roughly teardrop shaped, pointed anteriorly and rounded posteriorly, formed by the ectopterygoid laterally, pterygoid posteriorly, palatine anteromedially, and maxilla anterolaterally. The suborbital fenestra is broad in G. megalepis and the AB specimen and narrow in all the remaining specimens examined.

The epipterygoid is columnar, tilted posteriorly, and appears mildly medially bowed in G. megalepis, vertical or even externally bowed in G. humbloti, and vertical in G. maculata. It extends from the fossa columellae of the pterygoid toward, but not into contact with, the crista alaris of the prootic. The dorsal end is somewhat broader than the rest of the bone. The interpterygoid vacuity is hourglass shaped, but broadens more rapidly posteriorly than anteriorly, and is anteriorly bifurcated as a result of the medial spur of the vomer. The stapes has an oval footplate that fits in the fenestra ovalis, and two posts extending laterally from footplate, one anterior, the other posterior, converge to form the stapedial stem, leaving an open stapedial foramen. The fenestra ovalis is posterior to the quadrate.

The basioccipital underlies most of the braincase, is slightly wider than long, and lacks a distinct basal tubera. It is in contact with the parabasisphenoid anteriorly, otooccipitals laterally, and forms the ventral component of the foramen magnum. It is excluded from participation in the lateral aperture of the recessus scala tympani by the otooccipital. It is posterolaterally bordered by anteroventral extensions of the otooccipitals forming the sphenooccipital tubercle, which is connected to a sharp crista tuberalis.

The parabasisphenoid is in contact with the prootic dorsally and basioccipital posteriorly. It possesses a short, pointed parasphenoid rostrum, which is an extension of the squared anterior ends of the cristae trabeculae. The basipterygoid processes diverge anterolaterally, broadening distally, with flat, curved distal ends forming a synovial joint with the corresponding fossa of the posterior pterygoid (Payne, Holliday & Vickaryous, 2011). The vidian bridge extends to the base of the basipterygoid process from the crista prootica of the prootic. Posteriorly, the crista sellaris forms the anterior wall of the sella turcica. It has two pairs of anterior openings: carotid canals opening anteromedially, and the anterior openings of the Vidian canal anterolaterally, parallel to the basipterygoid processes.

The supraoccipital contacts the prootic anteriorly and otooccipitals ventrally, forming the dorsal edge of the foramen magnum. The posterior semi-circular canal extends posteriorly to the dorsal margin of the foramen magnum. The supraoccipital has a pair of dorsal tubercles that project dorsally without contacting the parietal; these tubercles are on either side of the midline of the holotype of G. megalepis, but are not strongly raised in any other specimen examined.

The prootic is thin and has a prominent, triangular crista alaris. It is in contract with the descending parietal process dorsally, the parabasisphenoid anteroventrally, supraoccipital posterodorsally, otooccipital posteroventrally, and almost in contact with the epipterygoid at the end of the crista alaris. The posterolateral margin forms the anterior wall of the fenestra ovalis, and the posteromedial surface forms the anterolateral wall of the brain case. The anterior semi-circular canal runs through the base of the alary process and crista alaris. The horizontal semicircular canal and the ampullar bulge are visible in the posterior edge of the prootic. A projection from the crista alaris extends anteromedially down to the crista sellaris of the sphenoid and contains the trigeminal foramen (Daza, Bauer & Snively, 2013), flaring also anterolaterally to the level of the epipterygoid from the base of the crista alaris.

The otooccipital is in contact with the prootic anteriorly, basioccipital ventromedially, supraoccipital dorsally, and the squamosal on the anterior face of the distal end of the paroccipital process. The horizontal and posterior semi-circular canals are visible as a bulge in posterior view. The occipital recess is enclosed in its posteroventral face. Anterodistally it projects ventrally to participate in the sphenooccipital tubercle with the basioccipital. The paroccipital process is long and thin, but broad dorsoventrally.

The foramen magnum is suboval, formed by the supraoccipital dorsally, otooccipitals lateroventrally, and basioccipital ventrally. The occipital condyles are double, formed by the otooccipitals laterally and the basioccipitals medially.

Jaw (Figs. 6D and 6E): The jaw curves medially anteriorly, and forms a weak symphysis with its counterpart. The dentary is the longest bone, being tubular and enclosing the Meckelian canal, which becomes broader posteriorly, as it approaches the mandibular fossa. It bears isodont, pleurodont, somewhat conical and some recurved teeth. Tooth loci varies considerably from 27 to 40, the smallest number on the left ramus of the AB specimen, but it seems that this specimens has a pathological condition since the number is higher on the other side, reaching the base of the coronoid eminence. The interdental space is larger in the AB specimen, which also explains the lowest number. Maximum tooth loci is roughly 34 in G. maculata, 37 in G. megalepis, and 40 in G. humbloti; with some teeth clearly missing in all specimens). Five mental foramina are present in the holotype, six in the paratype ZSM 289/2004 of G. megalepis. Posteriorly, the dentary contacts the surangular portion of the compound bone by superior and ventral processes, the latter extending considerably further than the former. The Meckelian canal is not outwardly pronounced, and opens anteriorly below the symphysis.

The splenial is a thin, triangular, and flat bone that forms the medial wall of the Meckelian canal, posterior to the tubular portion. It has two discrete foramina, the anterior inferior dental foramen and the anterior mylohyoid foramen.

The coronoid has a strong and fin-like dorsal eminence with a broadened anterior edge, but its precise shape varies within species. It inserts into the dentary at the level of the last or penultimate tooth (except on one side of the jaw of the AB specimen, as mentioned above). The posteromedial process reaches the middle of the surangular, anterior to the distinct mandibular fossa. The triangular splenial is present on the lingual surface of the mandible, in contact with the posteromedial surface of the dentary, the lingual anteroventral face of the coronoid, and the lingual surface of the surangular.

The dorsal edge of the surangular portion of the compound bone is concave. The Meckelian canal is closed, extending into the dentary from the adductor fossa. Surangular and posterior surangular foramina are located in the labial side of the compound bone. An external foramen for the chorda tympani is present at the base of the retroarticular process of the compound bone. The length, width, and concavity of the retroarticular process are variable within species probably due to scaling of the jaw muscles. The retroarticular process is strongly laterally notched, with a medial ridge on its articular surface.

Axial Skeleton (Fig. 7): 26 presacral, two sacral, and a varying number of caudal vertebrae are present (the total number cannot be ascertained due to autotomized or regenerated tails in all scanned specimens). Of the presacrals, eight are cervical (defined as being anterior to the first vertebra possessing a rib reaching the sternum), sixteen or seventeen are thoracic (rib-bearing), and one or two lack ossified ribs and are thus considered lumbars.

The atlas has an unfused neural arch, which is also not fused to the centrum, each side with a short dorsolateral posterior projection not overlying the axis. The odontoid process of the axis extends forward between the walls of atlas and into the braincase, fitting in between the occipital condyles. The anterior three cervical vertebrae (atlas, axis, and third cervical) lack ribs. The following five bear ribs of increasing length, all of which are to some degree dorsoventrally broadened.

The vertebrae are notochordal amphicoelous type (Romer, 1956). The ribs of the first four thoracic vertebrae reach the sternum—the fourth via the xiphisternum—followed by seven vertebrae articulating with long, posteriorly arching ribs distally associated with postxiphisternal inscriptional ribs, followed by five or six vertebrae possessing shorter ribs gradually becoming more posteriorly curved (see Fig. 7B); one or two lumbar vertebrae follow that are similar in shape to the posterior thoracic vertebrae but lack ribs.

The sacral pleurapophysis of the first sacral vertebra juts slightly posteriorly, articulating distally with the pelvic girdle. The posterodistal edge is fused to the anterior edge of the pleurapophysis of the second sacral vertebra, forming the foramen sacrale. The second sacral vertebra possesses a dorsoventrally thin posterior crest comprising almost half the distal breadth of the pleurapophysis (with asymmetrically emarginated distal edges in G. megalepis specimens: more emarginated on the left than the right in ZSM 2126/2007 and right than left in ZSM 289/2004);

The first five caudal vertebrae possess long thin lateral processes, initially extending beyond the sacrals, gradually decreasing in breadth, jutting posterolaterally, straight in G. maculata and G. humbloti, curved laterally in G. megalepis, becoming increasingly posterior-jutting. The first three caudals lack hemal arches.

Pectoral Girdle (Fig. 8): The pectoral girdle is comprised of paired clavicles, epicoracoids, and scapulocoracoids, and a non-paired interclavicle and presternum.

The presternum is kite-shaped, and varies in ossification levels from poorly to fully ossified. It has a synchondrotic articulation with the first three sternal ribs along its posterolateral border, but lacks distinct facets for these. Its anterolateral edges are thickened to form the coracosternal groove. No frontanelles are present. The mesosternal extension of the xiphisternum is variably long, but poorly ossified.

The sagittal interclavicle is posteriorly arrowhead-shaped, and extends less than one third into the sternum. It is anteriorly elongated, tubular and tapering, extending between the clavicles but not beyond them.

The suprascapular and epicoracoid regions are at least partly ossified, but never completely. The scapulocoracoid is typical in being composed of a horizontal plate (coracoid portion) and a vertical lateral ascending process (scapular portion). No clear suture of the scapula and coracoid is visible in the micro-CT scans. The coracoid portion is broad, plate-like, with a bulbous process at its posterolateral corner. The scapular portion is long, at least as long as the coracoid portion. Three rays are present, all of which are directed anteromedially: the scapular ray is slender, and passes dorsally anterior to the ascending lateral process of the clavicle in G. maculata, but does not surpass the clavicle in G. megalepis or G. humbloti—it defines the dorsal edge of the scapulocoracoid fenestra, which is ventrally completed by the primary coracoid ray. The secondary coracoid ray extends to the level of the clavicular fenestra in G. maculata, and to the level of the posterior margin of the clavicular fenestra in G. megalepis and G. humbloti. The rays define four fenestrae: the secondary coracoid fenestra (medial scapulocoracoid + secondary coracoid ray); the primary coracoid fenestra (secondary coracoid ray + primary coracoid ray); the scapulocoracoid fenestra (primary coracoid ray + scapular ray); and the scapular fenestra (scapular ray+distal scapulocoracoid). Anteriorly, all four fenestrae are closed by the cartilaginous epicoracoids, which are not rendered in our micro-CT scans (see Fig. 8). This formation is type 6 sensu Lécuru (1968). The supracoracoid foramen is small, lying closer to first coracoid fenestra than to the glenoid fossa.

The clavicle curves posteriorly and dorsally from the midline. It articulates with the ossified acromion process of the poorly ossified suprascapula. It is angled posterolaterally, with a broadly expanded but dorsoventrally flat medial portion—containing a large, oblong clavicular fenestra—and slender curving lateral portion. It articulates at the midline with its contralateral and the interclavicle, and is dorsally exceeded or at least overlapped by the epicoracoid cartilage and parts of the suprascapular rays.

Forelimbs (Fig. 8): The humerus is marginally longer than the radius and ulna. It is somewhat sigmoidal in dorsal view, with expanded proximal and distal ends. The proximal end is slightly less broad than the distal end. It possesses prominent humeral and deltopectoral crests (the latter with a sharp break separating it from the rest of the proximal humerus dorsally), as well as a moderately developed ectepicondylar crest and ectepicondyle. The bicipital fossa is deeply concave. The ectepicondylar foramen is visible in posterior view. In summary, it is fairly typical of gekkonids (Russell & Bauer, 2008).

The radius is long and thin, slightly dorsoventrally flattened and weakly curved, with its distal articulatory facet with a distinct processus styloideus; its distal end articulates with the radiale posteriorly. The ulna is slender, dorsoventrally flattened, and straight, narrowing distally, but flaring at its distal end, where it articulates with the ulnare laterally and pisciform ventrally. The olecranon process is clearly distinct, and proximal to it, on the articular surface of the humerus, lies the sesamoid patella ulnaris, which is rounded. The internal face of the olecranon process forms a smooth sigmoid notch.

The spatium interosseum is formed by the diverging radius and ulna proximally and the ulnare, centrale, and radiale distally, rendering it roughly teardrop shaped.

Nine carpal elements are present: the ulnare and radiale are subequal in size. The centrale is thin and long, and lies between these two elements. The pisciform is small and rounded, lies below the ulnare, and is probably not a true carpal (Russell & Bauer, 2008). A further five distal carpals are identifiable, the first in contact with phalange I, second with phalange II, third with phalanges II and III, fourth with phalanges III and IV, and fifth with phalange V. The phalangeal formula is 2-2-3-4-2.

The first phalange of the first finger, first of the second finger, first and second of the third finger, first through third of the fourth finger, and first of the fifth finger, are dorsoventrally flattened and laterally broadened. The terminal phalanges of each digit are slender and arcuate, ending in a laterally compressed, square tip with a distal claw-like projection, underlying the claws proper. The digits are able to hyperextend significantly. These three characters are presumably related to the adhesive pads of the fingers (Russell & Bauer, 2008). No ossified paraphalangeal elements are present in the micro-CT scans.

Pelvis (Fig. 9): The pelvis is composed of fused paired ilia, ischia, and pubes. The ischiopubic fenestra formed by the ischia and pubes is cardioid in shape, anteriorly rounded at the medial symphysis of the pubes in G. maculata, but more pointed in G. megalepis and G. humbloti—this fenestra may be medially divided by a proischiadic cartilage, but only the posterior-most portion of this element is shown in our micro-CT scans.

The pubis and ischia are broad and thin, concave in ventral view. The ilium is long, dorsoventrally broadened but laterally thin, and curves posteriorly. In lateral view (Fig. 9B), the iliac blade is reminiscent of the shoes of the Greek god Hermes—it rises posterodorsally, and is in broad medial contact with the pleurapophysis of the first sacral, and brief medial contact with the anterior portion of the pleurapophysis of the second sacral. Anterolaterally the ilium participates in the dorsal portion of the acetabulum.

The epipubic cartilage anterior to the medial pubic symphysis is somewhat calcified.

The pubis curves from the anterior acetabulum ventrally and medially to the anterior symphysis with its contralateral at the front of the pelvis. It has a strong, medioventrally jutting pubic tubercle on the posterior portion of its lateral edge (see p. 138–145 in Russell & Bauer (2008) for discussion of terminology), which descends more or less vertically in G. maculata and G. humbloti, but is posteriorly curved in G. megalepis. Medial to this is the concavo-convex pubic apron, the anterolateral edge of which runs anteromedially toward the medial symphysis with the contralateral pubis. The relatively large obturator foramen lies posterior to the pelvic tubercle, in line with the medial edge of the acetabulum.

Dorsolaterally, the ischium forms the posterior margin of the acetabulum. Ventromedially, it is roughly equal in breadth to the pubis, broadening toward the posteromedial symphysis with its contralateral at the back of the girdle. It possesses an almost lateral–pointing ischiadic tuberosity, rendering the posterolateral margin of the ischium deeply concave. The anterior margin of the ischium is also concave, extending anteriorly to form a weak prong, associated with the proischiadic cartilage. The medial ischial symphysis is not strongly fused, and the cartilaginous hypoischium likely extends into it.

A pair of curved post-cloacal bones is present in one specimen (ZSM 2126/2007; Fig. 9), but is absent from all other specimens. These may be sexually dimorphic and/or age dependent; see Russell, Vickaryous & Bauer (2016) for a review of their phylogenetic distribution and evolution.

Hindlimbs (Fig. 9): The femur is slender and weakly sigmoidal, with broad terminal ends. The epiphyseal internal trochanter is strong, and lies distal to the femoral condyle, from which it is separated by a deep notch. Its shape differs slightly among species: in G. megalepis, it is distinct and bulbous, in G. maculata it is ridge-like without a bulbous end, and in G. humbloti it is variable, with one specimen (ZSM 81/2006) resembling G. megalepis and one (ZSM 80/2010) resembling G. maculata. The ventral face of the proximal end of the femur has a moderately deep intertrochanteric fossa. The lateral distal condyle is distinctly larger than the medial one. The distal intercondylar groove is pronounced, and the popliteal fossa is not strongly deepened.

The fibula articulates via a sesamoid cyamella (=parafibula) with the lateral surface of the posterior femoral condyle. Additional sesamoids include the lenticular tibial patella dorsal to the distal end of the femur, a spherical post-axial ligament sesamoid (fabella), and the tibial lunula between the tibia and femur.

The fibula is laterally flattened to a slender rod of bone, with a slightly broader distal than proximal end. The tibia is broad and dorsoventrally compressed, and bows slightly outward. The cnemial and ventral crests are not pronounced and may be absent. Together the tibia and fibula articulate with the subtriangular astragalocalcaneum, which in turn articulates with the first metatarsal and the fourth distal tarsal Two distal tarsals are present (third and fourth; see Russell & Bauer (2008) for discussion of terminology). The fifth metatarsal is L-shaped and bears distinctly raised anterior and posterior plantar tubercles. The second metatarsal is longer than the fourth, and the third is the longest. The pedal phalangeal formula is 2-2-3-4-3.

Discussion

Geckolepis megalepis is the first Geckolepis species to be described in 75 years (and it has been 123 years since the last currently recognised species was described). Although far northern Madagascar is relatively well surveyed for reptiles, and numerous Geckolepis populations have been recorded from this area, the new species has not been found outside the Ankarana massif. Considering the increasing number of squamate species putatively endemic to this spectacular limestone formation (Glaw et al., 2010; Glaw et al., 2012; Jono et al., 2015; Ruane et al., 2016), it is likely that G. megalepis will also turn out to be microendemic to the region; unusual among Geckolepis species. In Ankarana, it occurs in sympatry with another lineage of Geckolepis (OTU G in Lemme et al., 2013), which also may be microendemic to this area, but to which it is only distantly related (uncorrected pairwise distance 11.3% in 12S rRNA according to sequences published by Lemme et al. (2013). The new species can be distinguished from these geckos by the lack of a dark stripe from eye to ear opening (vs. presence), possession of 17–18 scale rows at midbody (vs. 25–28), and possession of 27–31 ventral scale rows (vs. 33–43) (Lemme et al., 2013).

Extreme integumentary autotomy in Geckolepis megalepis

Many reptiles have evolved the ability to shed some part of their body in response to predator attack. The most widespread form is caudal autotomy, the shedding of all or part of the tail, which is widespread among Lepidosauria, from amphisbaenians to rhynchocephalians, even being found in some snakes (Arnold, 1984; Bateman & Fleming, 2009). Geckolepis species are also able to shed their tails, and indeed few specimens survive to adulthood with their original tails intact (see for instance Figs. 3A and 3B). In addition, these geckos have evolved an even more extreme adaptation, i.e. the autotomy of virtually their entire integument when seized or even touched. Earlier studies have shown that the autotomized layers include epidermis, underlying connective tissue, and subcutaneous fat tissue, and that a layer between the integument and the underlying tissue represents a pre-formed splitting zone (Schubert & Christophers, 1985). The shedding process is most likely achieved by contraction of the network of myofibroblasts in the preformed splitting zone, with vasoconstriction in the most superficial vasculature of the dermis to avoid bleeding (Schubert & Christophers, 1985). This process is thus completely different from the normal skin shedding of squamate reptiles, which leads to a loss of keratinized epidermis only (Schubert & Christophers, 1985) The scarless regeneration of the whole integument occurs within a few weeks, apparently starting from stem cells of the deeper layers of the connecting tissue and is considered as unique among vertebrates (Schubert, Steffen & Christophers, 1990). Superficially, no differences are apparent between regenerated and original scales, due to the irregularity of scalation patterns and some variability in scale size. The same is true for regenerated tails; indeed, it is often hard to be certain that a Geckolepis tail has been regenerated without X-ray images showing that the vertebrae are absent.

The new species is remarkable in the possession of proportionally larger scales than any of its congeners (especially in the dorsal cervical region, see Figs. 3 and 4). Midbody dorsal scales of Geckolepis megalepis are 7.3–8.3% of the SVL (by comparison, a typical midbody dorsal scale of the holotype of G. maculata is just 5.4% of its SVL). Indeed as far as we have been able to determine, G. megalepis has the largest mid-body scales of any gecko in both relative and absolute terms, as its scales outstrip those of all known congeners, and only Teratoscincus may approach Geckolepis in scale size. Remarkably, the latter genus has similarly fragile skin (Bauer, Russell & Shadwick, 1993), which may have evolved for similar antipredatory function.

The exceptionally large scales of G. megalepis lead to questions about the possible advantage of larger scales for species with autotomizable integument. As is visible in Fig. 3C, the large, imbricate body scales of Geckolepis megalepis are attached to the integument only superficially by a narrow transparent zone which covers less than 20% of the scale’s edge surface. Schmidt (1911) called this region the ‘Anwachsfläche’ (literally ‘attachment area’), and noted that it is much smaller relative to the size of the scales of Geckolepis than in other lizards. With increasing scale diameter, the circumference of the scale and therefore its zone of connectivity increases linearly while the area of the scales increases exponentially (approximating the scales to a circular shape). This increases the surface area and therefore force of friction on scales exponentially while the ‘tear zone’ of the scales increases linearly, meaning that there is a smaller tear zone per unit area with greater scale size. Thus, less force, applied in a posterior or lateral direction, should be required to remove a larger scale than a smaller one. Additionally, as the scales are imbricate, leverage may play a role: an anteriorly directed pressure on the scales may cause them to lift and detach, much as one might detach a sticky note from a surface. The leverage of a longer scale is greater than a shorter one, and these may therefore lift more easily. This is probably further enhanced by the steeper angle of larger scales to the body (Schmidt, 1911). Together, these principles may explain why the scales of G. megalepis appear to come off more easily than those of other Geckolepis species, but further studies are clearly necessary to confirm or reject this hypothesis.

Although it is highly plausible to interpret their ability of scale autotomy as an anti-predator defence mechanism, direct observations of predation events or attempts on Geckolepis individuals are scarce and include only a scorpion (Grosphus flavopiceus), a bird (Dicrurus forficatus) and a large nocturnal Blaesodactylus gecko (Gardner & Jasper, 2014; Gardner & Jasper, 2015; Glaw, Vences & Lourenco, 2002). In the lattermost case, the Geckolepis individual slipped from the mouth of the Blaesodactylus ca. 30 s after being captured, and escaped denuded (Gardner & Jasper, 2015), thereby providing the first direct evidence of successful escape by skin shedding. Further studies on the role of dermolytic scale autotomy by Geckolepis are clearly needed, in order to understand its functionality with a range of predators (its function against snakes, for instance, has not yet been observed), and to understand the pressures driving the evolution of greater scale size in this particular lineage of Geckolepis.

The osteology of Geckolepis and the next steps in resolving their taxonomy

Our osteological description of the genus Geckolepis, based on six specimens of four species, reveals strong morphological conservatism in this genus. Few characters show potentially diagnostic differences at the species level (see Appendix S2), and the degree of intra-specific variation is apparently quite high. However, through the use of micro-CT, we were able to include the holotype of G. maculata in our skeletal description. This will be an important step toward the resolution of its identity, despite our continued failure to trace its type locality (see Supplemental Information). Lemme et al. (2013) assigned their OTU AB to this species on the basis of its overall similar morphology, but our analysis of their skulls suggests that they are probably not conspecific. This means that the assignment of this name is still completely unknown; it belongs either to another of the known clades, or to one not yet characterised. A broader survey of the osteology of the genus will be required to resolve the identity of this species, and will in turn yield the total resolution of this genus.

Despite the detailed osteological description, we admit that Geckolepis represents an extremely difficult taxonomic group that is hard to characterise. The high variability in scale number and enhanced ability to shed scales upon capture has misled taxonomists in the past into believing they were dealing with distinct new species, which have subsequently been synonymised (Angel, 1942; Köhler et al., 2009). The trouble is further enhanced by multiple genetic lineages occurring in sympatry (Lemme et al., 2013; Fig. 1), and further still by apparent osteological conservatism. However, we were able to show that this is not always the case, and members of the AB OTU of Lemme et al. (2013) for instance show strong osteological differences that will facilitate its description. Nevertheless, the osteology did not provide as many taxonomic characters as we had hoped. Admittedly our sample size is small, and therefore practically no data yet exist on the degree of inter- and intraspecific osteological variation in these geckos. Examination of many further specimens and other lineages will enhance our ability to use osteology as a source of characters in their taxonomic resolution.

Thus, the next steps are now clear: (1) a survey of osteology in this genus in the context of molecular phylogenetic relationships of OTUs, and (2) a survey of intraspecific and sexual skeletal variability in at least one lineage, although this is generally minimal among gekkotans (Daza et al., 2008; Daza et al., 2009). Based on this data and corresponding other datasets, we must establish with a high degree of certainty which OTU from Lemme et al. (2013) really corresponds to G. maculata (if any). Once this information is gathered, we may proceed with the resolution of the taxonomy of the genus.

The framework of an integrative dataset composed of morphological, meristic, molecular phylogenetic, and osteological data has considerable potential for dealing with species complexes in squamates—even those as tortuous as Geckolepis, and certainly in other species with fragile skin. However, it is clear that the robustness of conclusions strongly depends on the available sample size. In instances, such as this one, where sample size is limited to a low number of specimens, any osteological, morphological, or pholidotic feature identified as differing must first be highlighted as being potentially diagnostic, until more data becomes available to verify the value of each of these features. Nevertheless, the value of these data, especially when they are extracted from holotypes and old specimens in a non-destructive way, cannot be overstated. Micro-CT is therefore likely to have a pivotal role in resolving many difficult species complexes.

Supplemental Information

Video S1 Rotational video of the skull of Geckolepis megalepis sp. nov. (ZSM 2126/2007)

Click here for additional data file.

Video S2 Rotational video of the skull of Geckolepis maculata (ZMB 9655)

Click here for additional data file.

Video S3 Rotational video of the skull of Geckolepis of OTU AB (ZSM 1520/2008)

Click here for additional data file.

Video S4 Rotational video of the skull of Geckolepis humbloti (ZSM 80/2010)

Click here for additional data file.

Video S5 Rotational video of the skull of Geckolepis humbloti (ZSM 81/2006)

Click here for additional data file.

Supplemental Information 1 Supplementary information

The type locality of Geckolepis maculata Peters, 1880

Click here for additional data file.

Appendix S1 Measurements and meristics of the specimens examined for this study

Click here for additional data file.

Appendix S2 Variable osteological characters among specimens of Geckolepis examined.

Click here for additional data file.

Figure S1 Stereoscopic 3D renderings of the skull of the holotype of Geckolepis megalepis sp. nov. in anterolateral and dorsal view

Click here for additional data file.

We thank the Malagasy authorities for issuing collection and export permits. Thanks are also due to O Hawlitschek, EZ Lattenkamp, H Rösler and CY Wang-Claypool for their support and help. P Bora, N D’Cruze, H Enting, M Franzen, K Glaw, T Glaw, Jaques, A Knoll, M Puente, NA Raharinoro and R Randrianiaina helped conduct fieldwork and Y Pareik with logistics and his great hospitality. We also thank N Holovacs for his assistance with the rotational videos, A Anker for permission to publish his photograph, and M-O Rödel for the loan of the G. maculata holotype. We thank D Paluh, A Griffing and AM Bauer for access to their manuscript on the osteoderms of Geckolepis. J Grismer and E Stanley gave helpful suggestions for the improvement of this article, and we are grateful to them.

Additional Information and Declarations

Competing Interests

Author Contributions

Animal Ethics

Field Study Permissions

Data Availability

New Species Registration

The authors declare there are no competing interests.

Mark D. Scherz conceived and designed the experiments, analyzed the data, contributed reagents/materials/analysis tools, wrote the paper, prepared figures and/or tables, reviewed drafts of the paper.

Juan D. Daza analyzed the data, contributed reagents/materials/analysis tools, wrote the paper, prepared figures and/or tables, reviewed drafts of the paper.

Jörn Köhler and Miguel Vences analyzed the data, contributed reagents/materials/analysis tools, wrote the paper, reviewed drafts of the paper.

Frank Glaw conceived and designed the experiments, analyzed the data, contributed reagents/materials/analysis tools, wrote the paper, reviewed drafts of the paper.

The following information was supplied relating to ethical approvals (i.e., approving body and any reference numbers):

No experiments were conducted on living vertebrates.

The following information was supplied relating to field study approvals (i.e., approving body and any reference numbers):

All field research and collecting of specimens were approved by the Malagasy Ministère de l’Environnement, des Eaux et des Forêts (Direction des Eaux et Forêts, DEF) under the following permits: 238-MINENVEF/SG/DGEF/DPB/SCBLF dated 14 November 2003; 298/06-MINENV.EF/SG/DGEF/DPB/SCBLF/RECH dated 22 December 2006; 036/08 MEEFT/SG/DGEF/DSAP/SSE dated 30 January 2008; 174/16/MEEF/SG/DGF/DSAP/SCB, dated 25 July 2016. Export of specimens was approved by the DEF under permits: 094C-EA03/MG04, dated 1 March 2004; 051N-EA03/MG07, dated 10 March 2007 and 270N-EA09/MG16, dated 7 September 2016.

The following information was supplied regarding data availability:

The micro-CT scan data has been deposited in MorphoSource: http://morphosource.org/Detail/ProjectDetail/Show/project_id/302.

The following information was supplied regarding the registration of a newly described species:

Geckolepis megalepis LSID: urn:lsid:zoobank.org:act:08097CA5-172E-4F45-AE68-AC4E02BBC644.

Publication LSID: urn:lsid:zoobank.org:pub:12C9B971-CF0F-49B7-93B1-0A480ACA9B53.

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
