# Peer review of "Off the scale: a new species of fish-scale gecko (Squamata: Gekkonidae: Geckolepis) with exceptionally large scales"

_PeerJ, doi:10.7717/peerj.2955_

## Round 0.1 · original submission · Minor Revisions

Dear Authors,

Thank you for submitting a fine MS to PeerJ. Both referees suggested minor revision, and I am also recommending minor revision.

I do not see any major issue that needs to be addressed. I also agree with the first reviewer that it would be easier for readers if they had access to a phylogeny in this paper. So I suggest that a phylogeny showing the OTUs of Lemme et al. (2013), but also including information from Hawlitschek et al (2016), be provided in this paper. I suggest collapsing individuals sequences into OTUs and assigning species names to OTUs if they exist and showing their geographic distribution. I think this would make it much easier on the reader to situate themselves and would graphically represent the motivation of this study.

Also it would be important to deposit the image data in a public database.

Otherwise a job well done, and I look forward to receiving your revision shortly.

Sincerely,

Tomas Hrbek

·

Basic reporting

The authors describe a new species of Geckolepis from northwest Madagascar that had been shown to be phylogenetically independent in a previous study. Not only is the holotype in great condition (especially for Geckolepis), but they provide a though examination of it and its comparison to other species in the genus for future studies. This paper was a great mixture of old school taxonomy with modern morphological imagining techniques and I thought it was awesome!

Experimental design

Excellent.

Validity of the findings

Valid. However, they need to show phylogenetic placement of G. megalepis with a distribution map of the new species in Madagascar. They may have provided one but I couldn't find it.

Additional comments

Great piece of work on a really cool group of geckos! Here are a few comments meant to help improve the overall quality of the paper. Contact me if you like.

Minor Comments
- On the figures comparing the different skulls of the species I think it would help if the diagnostic differences were colored so they stand out as soon as you see the figure.
-Lines 58-60 are a bit hard to follow and may need to be rewritten.
-This is a great paper because it also provides an example on how to approach similar questions in other geckos groups that have tear away skin, like Gehyra. I think you could add a sentence in the discussion about this and how this combination of old and new techniques could be beneficial for other groups.
- Get rid of the fill-in pattern on the gecko in Fig 1b. Simple outline of the lizard would look better.

Major Comments
- Many sections of this paper (ie lines 58-73) refer to Fig. 2 of Lemme et al. (2013) and the placement of OUT D in that phylogeny, and this phylogenetic position is one many line of evidence that indicates this lineage is a new species. The sections that make reference to this phylogeny in the text are hard to follow without the tree and I feel if there was a some version of fig 2 form Lemme et al. (2013) that would make these section easier to follow for the reader. I feel anytime you can show a species phylogenetic position in the same paper that it is described it always make the paper better and helps future taxonomic custodial work.

·

Basic reporting

See below

Experimental design

See below

Validity of the findings

See below

Additional comments

This paper, describing a new species of fish scaled gecko based on previous genetic work by some of the authors and a combination of traditional and CT-based morphology, is a well written, beautifully illustrated and interesting manuscript. The detailed skeletal morphological descriptions are particularly interesting, and provide an excellent example of the utility of CT for extracting novel data from rare and important specimens. I have a few, general comments and some small suggestions that may improve the paper. Firstly I think that an additional figure early in the paper, showing the locality and phylogenetic relationships of the Geckolepis complex, may help strengthen the introduction and help frame the phylogenetic and biogeographic context in the description- perhaps this could incorporated a modified phylogeny combined with a map of the sampled localities and type localities. I would also like to see a discussion of the size and morphology of the regenerated body scales and/or scales on regenerated tales. Thirdly, it may be constructive to discuss the ossification of the scales- I am aware of a paper by other authors (Paluh and Bauer) that is, or soon will be, in press that addresses this point in great detail, but a brief discussion of why these scales might be mineralized, and how this could potentially augment the effective shedding of skin, (citing Paluh and Bauer’s findings) could add an additional angle of interest to this paper. I have added number of smaller suggestions to the main text in a word document that is attached here. The figures are good, though I would make the scale bar in figure 4 smaller: 10mm would be more suitable.

Finally, I would suggest that the authors, in the spirit of open access, make their datasets (mesh files and CT slice data) available online through a morphological database like www.morphosource.org or an equivalent.


Ed Stanley

---

## Round 0.2 · accepted · Accept

Dear Authors,

I read through your revised version, and I am happy to recommend Acceptance. Thank you also for including Figure 1.

Congratulations on a fine study.

Cheers,

Tomas Hrbek